# Multi-modal Self-Supervision from Generalized Data Transformations

## Abstract

In the image domain, excellent representations can be learned by inducing invariance to content-preserving transformations, such as image distortions. In this paper, we show that, for videos, the answer is more complex, and that better results can be obtained by accounting for the interplay between invariance, distinctiveness, multiple modalities, and time. We introduce Generalized Data Transformations (GDTs) as a way to capture this interplay. GDTs reduce most previous self-supervised approaches to a choice of data transformations, even when this was not the case in the original formulations. They also allow to choose whether the representation should be invariant or distinctive w.r.t. each effect and tell which combinations are valid, thus allowing us to explore the space of combinations systematically. We show in this manner that being invariant to certain transformations and distinctive to others is critical to learning effective video representations, improving the state-of-the-art by a large margin, and even surpassing supervised pretraining. We demonstrate results on a variety of downstream video and audio classification and retrieval tasks, on datasets such as HMDB-51, UCF-101, DCASE2014, ESC-50 and VGG-Sound. In particular, we achieve new state-of-the-art accuracies of 72.8% on HMDB-51 and 95.2% on UCF-101.

## 1 Introduction

Recent works such as PIRL (Misra & van der Maaten, 2020), MoCo (He et al., 2019) and SimCLR (Tian et al., 2019) have shown that it is possible to pre-train state-of-the-art image representations without the use of any manually-provided labels. Furthermore, many of these approaches use variants of noise contrastive learning (Gutmann & Hyvärinen, 2010). Their idea is to learn a representation that is *invariant* to transformations that leave the meaning of an image unchanged (e.g. geometric distortion or cropping) and *distinctive* to changes that are likely to alter its meaning (e.g. replacing an image with another chosen at random).

An analysis of such works shows that a dominant factor for performance is the choice of the transformations applied to the data. So far, authors have explored ad-hoc combinations of several transformations (e.g. random scale changes, crops, or contrast changes). Videos further allow to leverage the time dimension and multiple modalities. For example, Arandjelovic & Zisserman (2017); Owens et al. (2016) learn representations by matching visual and audio streams, as a proxy for objects that have a coherent appearance and sound. Their formulation is similar to noise contrastive ones, but does not quite follow the pattern of expressing the loss in terms of data transformations. Others (Chung & Zisserman, 2016; Korbar et al., 2018; Owens & Efros, 2018) depart further from standard contrastive schemes by learning representations that can tell whether visual and audio streams are in sync or not; the difference here is that the representation is encouraged to be distinctive rather than invariant to a time shift.

Overall, it seems that finding an optimal noise contrastive formulation for videos will require combining several transformations while accounting for time and multiple modalities, and understanding how invariance and distinctiveness should relate to the transformations. However, the ad-hoc nature of these choices in previous contributions make a systematic exploration of this space rather difficult.

In this paper, we propose a solution to this problem by introducing the Generalized Data Transformations (GDT; fig. 1) framework. GDTs reduce most previous methods, contrastive or not, to a noise contrastive formulation that is expressed in terms of data transformations only, making it

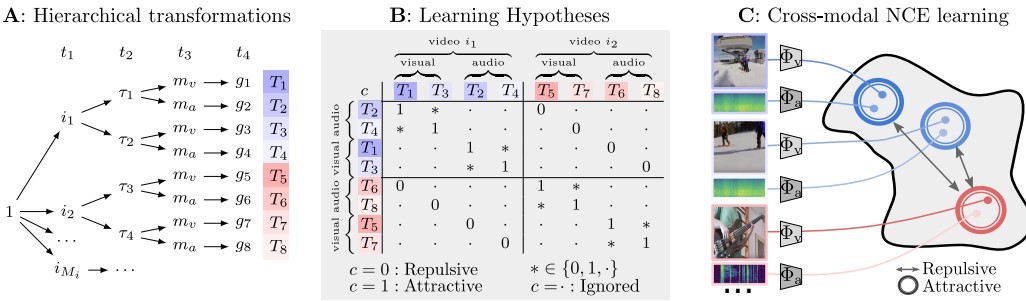

Fig. 1: **Schematic overview of our framework. A**: Hierarchical sampling process of generalized transformations $T = t_M \circ ... \circ t_1$ for the multi-modal training study case. **B**: Subset of the $c(T, T')$ contrast matrix which shows which pairs are repelling (0) and attracting (1) (see text for details). **C**: With generalized data transformations (GDT), the network learns a meaningful embedding via learning desirable invariances and distinctiveness to transformations (realigned here for clarity) across modalities and time. The embedding is learned via noise contrastive estimation against clips of other source videos. Illustrational videos taken from YouTube (Google, 2020).

simpler to systematically explore the space of possible combinations. This is true in particular for multi-modal data, where separating different modalities can also be seen as a transformation of an input video. The formalism also shows *which* combinations of different transformations are valid and how to enumerate them. It also clarifies how invariance and distinctiveness to different effects can be incorporated in the formulation and when doing so leads to a valid learning objective. These two aspects allows the search space of potentially optimal transformations to be significantly constrained, making it amenable to grid-search or more sophisticated methods such as Bayesian optimisation.

By using GDTs, we make several findings. First, we find that using our framework, most previous pretext representation learning tasks can be formulated in a noise-contrastive manner, unifying previously distinct domains. Second, we show that just learning representations that are invariant to more and more transformations is *not* optimal, at least when it comes to video data; instead, balancing invariance to certain factors with distinctiveness to others performs best. Third, we find that by investigating what to be variant to can lead to large gains in downstream performances, for both visual and audio tasks.

With this, we are able to set the new state of the art in audio-visual representation learning, with both small and large video pretraining datasets on a variety of visual and audio downstream tasks. In particular, we achieve $95.2\%$ and $72.8\%$ on the standardized UCF-101 and HMDB-51 action recognition benchmarks.

## 2 RELATED WORK

**Self-supervised learning from images and videos.** A variety of pretext tasks have been proposed to learn representations from unlabelled **images**. Some tasks leverage the spatial context in images (Doersch et al., 2015; Noroozi & Favaro, 2016) to train CNNs, while others create pseudo classification labels via artificial rotations (Gidaris et al., 2018), or clustering features (Asano et al., 2020b; Caron et al., 2018; 2019; Gidaris et al., 2020; Ji et al., 2018). Colorization (Zhang et al., 2016; 2017), inpainting (Pathak et al., 2016), solving jigsaw puzzles (Noroozi et al., 2017), as well as the contrastive methods detailed below, have been proposed for self-supervised image representation learning. Some of the tasks that use the space dimension of images have been extended to the space-time dimensions of **videos** by crafting equivalent tasks. These include jigsaw puzzles (Kim et al., 2019), and predicting rotations (Jing & Tian, 2018) or future frames (Han et al., 2019). Other tasks leverage the temporal dimension of videos to learn representations by predicting shuffled frames (Misra et al., 2016), the direction of time (Wei et al., 2018), motion (Wang et al., 2019), clip and sequence order (Lee et al., 2017; Xu et al., 2019), and playback speed (Benaim et al., 2020; Cho et al., 2020; Fernando et al., 2017). These pretext-tasks can be framed as GDTs.

**Multi-modal learning.** Videos, unlike images, are a rich source of a variety of modalities such as speech, audio, and optical flow, and their correlation can be used as a supervisory signal. This

idea has been present as early as 1993 (de Sa, 1994). Only recently, however, has multi-modal learning been used to successfully learn effective representations by leveraging the natural correspondence (Alwassel et al., 2020; Arandjelovic & Zisserman, 2017; Asano et al., 2020a; Aytar et al., 2016; Morgado et al., 2020; Owens et al., 2016) and synchronization (Chung & Zisserman, 2016; Korbar et al., 2018; Owens & Efros, 2018) between the audio and visual streams. A number of recent papers have leveraged speech as a weak supervisory signal to train video representations (Li & Wang, 2020; Miech et al., 2020; Nagrani et al., 2020; Sun et al., 2019a;b) and recently Alayrac et al. (2020), which uses speech, audio and video. Other works incorporate optical flow and other modalities (Han et al., 2020; Liu et al., 2019; Piergiovanni et al., 2020; Zhao et al., 2019) to learn representations. In (Tian et al., 2019), representations are learned with different views such as different color channels or modalities) to induce invariances. In contrast, our work analyses multi-modal transformations and examines their utility when used as an invariant *or* variant learning signal.

**Noise Contrastive Loss.** Noise contrastive losses (Gutmann & Hyvärinen, 2010; Hadsell et al., 2006) measure the similarity between sample pairs in a representational space and are at the core of several recent works on unsupervised feature learning. It has been shown to yield good performance for learning image (Chen et al., 2020b; He et al., 2019; Hénaff et al., 2019; Hjelm et al., 2019; Li et al., 2020; Misra & van der Maaten, 2020; Oord et al., 2018; Tian et al., 2019; 2020; Wu et al., 2018) and video (Han et al., 2019; Li & Wang, 2020; Miech et al., 2020; Morgado et al., 2020; Sohn, 2016; Sun et al., 2019a) representations, and circumvents the need to explicitly specify what information needs to be discarded via a designed task.

We leverage the noise contrastive loss as a learning framework to encourage the network to learn desired invariance and distinctiveness to data transformations. The GDT framework can be used to combine and extend many of these cues, contrastive or not, in a single noise contrastive formulation.

## 3 METHOD

A *data representation* is a function $f : \mathcal{X} \to \mathbb{R}^D$ mapping data points $x$ to vectors $f(x)$. Representations are useful because they help to solve tasks such as image classification. Based on the nature of the data and the task, we often know a priori some of the invariances that the representation should possess (for example, rotating an image usually does not change its class). We can capture those by means of the *contrast* function[1] $c(x_1, x_2) = \delta_{f(x_1)=f(x_2)}$, where $c(x_1, x_2) = 1$ means that $f$ is invariant to substituting $x_2$ for $x_1$, while $c(x_1, x_2) = 0$ means that $f$ is distinctive to this change. Any partial knowledge of the contrast $c$ can be used as a cue to learn $f$, but $c$ is not arbitrary: in order for $c$ to be *valid*, the expression $c(x_1, x_2) = 1$ must be an *equivalence relation* on $\mathcal{X}$, i.e. be reflexive $c(x, x) = 1$, symmetric $c(x_1, x_2) = c(x_2, x_1)$ and transitive $c(x_1, x_2) = c(x_2, x_3) = 1 \Rightarrow c(x_1, x_3) = 1$. This is justified in Appendix A.1 and will be important in establishing which particular learning formulations are valid and which are not.

We introduce next our *Generalized Data Transformations* (GDTs) framework by generalizing two typical formulations: the first is analogous to 'standard' methods such as MoCo (He et al., 2019) and SimCLR (Chen et al., 2020b) and the second tackles multi-modal data.

**Standard contrastive formulation.** Recall that the goal is to learn a function $f$ that is compatible with a known contrast $c$, in the sense explained above. In order to learn $f$, we require positive ($c(x_1, x_2) = 1$) and negative ($c(x_1, x_2) = 0$) example pairs $(x_1, x_2)$. We generate positive pairs by sampling $x_1$ from a data source and then by setting $x_2 = g(x_1)$ as a random transformation of the first sample, where $g \in \mathcal{G}$ is called a data *augmentation* (e.g. image rotation). We also generate negative pairs by sampling $x_1$ and $x_2$ independently.

It is convenient to express these concepts via transformations only. To this end, let $D = (x_1, \ldots, x_N) \in \mathcal{X}^N$ be a collection of $N$ i.i.d. training data samples. A *Generalized Data Transformation* (GDT) $T : \mathcal{X}^N \to \mathcal{Z}$ is a mapping that acts on the set of training samples $D$ to produce a new sample $z = TD$. Note that the GDT is applied to the entire training set, so that sampling itself can be seen as a transformation. In the simplest case, $\mathcal{Z} = \mathcal{X}$ and a GDT $T = (i, g)$ extracts the sample corresponding to a certain index $i$ and applies an augmentation $g : \mathcal{X} \to \mathcal{X}$ to it, i.e. $TD = g(x_i)$.

---

[1] We use the symbol $\delta$ to denote the Kronecker delta.

Usually, we want the function $f$ to be *distinctive* to the choice of sample but *invariant* to its augmentation. This is captured by setting the contrast $c(T, T')^2$ to $c((i, g), (i', g')) = \delta_{i=i'}$. Given a batch $\mathcal{T} = \{T_1, \ldots, T_K\}$ of $K$ GDTs, we then optimize a pairwise-weighted version of the *noise-contrastive loss* (Chen et al., 2020b; Gutmann & Hyvärinen, 2010; Oord et al., 2018; Tian et al., 2019; Wu et al., 2018), the GDT-NCE loss:

$$\mathcal{L}(f; \mathcal{T}) = - \sum_{T, T' \in \mathcal{T}} c(T, T') w(T, T') \, \log \left( \frac{\exp \langle f(TD), f(T'D) \rangle / \rho}{\sum_{T'' \in \mathcal{T}} w(T, T'') \, \exp \langle f(TD), f(T''D) \rangle / \rho} \right). \quad (1)$$

Here, the scalar $\rho$ is a temperature parameter and the weights $w(T, T')$ are set to $\delta_{T \neq T'}$ in order to discount contrasting identical transformations, which would result in a weak learning signal. Minimizing eq. (1) pulls together vectors $f(TD)$ and $f(T'D)$ if $c(T, T') = 1$ and pushes them apart if $c(T, T') = 0$, similar to a margin loss, but with a better handling of hard negatives (Chen et al., 2020b; Khosla et al., 2020; Tian et al., 2019).[3] When using a single modality, $T = T'$ and positive pairs are computed from two differently augmented versions.

**Multi-modal contrastive formulation.** We now further extend GDTs to handle multi-modal data. In this case, several papers (Arandjelovic & Zisserman, 2017; Aytar et al., 2016; Korbar et al., 2018; Owens et al., 2016; Wei et al., 2018) have suggested to learn from the correlation between modalities, albeit usually not in a noise-contrastive manner. In order to encode this with a GDT, we introduce *modality projection* transformations $m \in \mathcal{M}$. For example, a video $x = (v, a)$ has a visual component $v$ and an audio component $a$ and we we have two projections $\mathcal{M} = \{m_a, m_v\}$ extracting respectively the visual $m_v(x) = v$ and audio $m_a(x) = a$ signals. We can plug this directly in eq. (1) by considering GDTs $T = (i, m)$ and setting $TD = m(x_i)$, learning a representation $f$ which is distinctive to the choice of input video, but invariant to the choice of modality.[4]

**General case.** Existing noise contrastive formulations learn representations that are invariant to an ad-hoc selection of transformations. We show here how to use GDTs to build systematically new *valid* combinations of transformations while choosing whether to encode invariance *or distinctiveness* to each factor. Together with the fact that all components, including data sampling and modality projection, are interpreted as transformations, this results in a powerful approach to explore a vast space of possible formulations systematically, especially for the case of video data with its several dimensions.

In order to do so, note that to write the contrastive loss eq. (1), we only require: the contrast $c(T, T')$, the weight $w(T, T')$ and a way of sampling the transformations $\mathcal{T}$ in the batch. Assuming that each generalized transformation $T = t_M \circ \cdots \circ t_1$ is a sequence of $M$ transformations $t_m$, we start by defining the contrast $c$ for individual factors as:

$$c(t_m, t_m') = \begin{cases} 1, & \text{if we hypothesize invariance,} \\ \delta_{t_m = t_m'}, & \text{if we hypothesize distinctiveness.} \end{cases} \quad (2)$$

The overall contrast is then $c(T, T') = \prod_{m=1}^{M} c(t_m, t_m')$. In this way, each contrast $c(t_m, t_m')$ is an equivalence relation and so is $c(T, T')$ (see Appendix A.1), making it valid in the sense discussed above. We also assume that $w(T, T') = 1$ unless otherwise stated.

Next, we require a way of sampling transformations $\mathcal{T}$ in the batch. Note that each batch *must* contain transformations that can be meaningfully contrasted, forming a mix of invariant and distinctive pairs, so they cannot be sampled independently at random. Furthermore, based on the definition above, a single 'distinctive' factor in eq. (2) such that $t_m \neq t_m'$ implies that $c(T, T') = 0$. Thus, the batch must contain several transformations that have equal distinctive factors in order to generate a useful learning signal.

A simple way to satisfy these constraints is to use a *hierarchical* sampling scheme (fig. 1) First, we sample $K_1$ instances of transformation $t_1$; then, for each sample $t_1$, we sample $K_2$ instances

---

[2]Note that, differently from the previous section, we have now defined $c$ on transformations $T$ rather than on samples $x$ directly. In Appendix A.1, we show that this is acceptable provided that $c(T, T') = 1$ also defines an equivalence relation.

[3]We can think of eq. (1) as a softmax cross-entropy loss for a classification problem where the classes are the equivalence classes $\mathcal{T}/c$ of transformations.

[4]For this, as $f$ must accept either a visual or audio signal as input, we consider a pair of representations $f = (f_v, f_a)$, one for each modality.

of transformation $t_2$ and so on, obtaining a batch of $K = \prod_{m=1}^{M} K_m$ transformations $T$. In this manner, the batch contains exactly $K_M \times \cdots \times K_{m+1}$ transformations that share the same first $m$ factors $(t_1 = t'_1, \ldots, t_m = t'_m)$. While other schemes are possible, in Appendix A.2.1, we show that this is sufficient to express a large variety of self-supervised learning cues that have been proposed in the literature. In the rest of the manuscript, however, we focus on audio-visual data.

### 3.1 EXPLORING CONTRASTIVE AUDIO-VISUAL SELF-SUPERVISION

Within multi-modal settings, video representation learning on audio-visual data is particularly well suited for exploring the GDT framework. Especially compared to still images, the space of transformations is much larger in videos due to the additional time dimension and modality. It is therefore an ideal domain to explore how GDTs can be used to limit and explore the space of possible transformations and their quality as a learning signal when used as variances or invariances. In order to apply our framework to audio-visual data, we start by specifying how transformations are sampled by using the hierarchical scheme introduced above (see also Figure 1). We consider in particular GDTs of the type $T = (i, \tau, m, g)$ combining the following transformations. The first component $i$ **selects** a video in the dataset. We sample $K_i \gg 2$ indices/videos and assume distinctiveness, so that $c(i, i') = \delta_{i=i'}$. The second component $\tau$ contrasts different **temporal shifts**. We sample $K_\tau = 2$ different values of a delay $\tau$ uniformly at random, extracting a 1s clip $x_{i\tau}$ starting at time $\tau$. For this contrast, we will test the distinctiveness and invariance hypotheses. The third component $m$ contrasts **modalities**, projecting the video $x_{i\tau}$ to either its visual or audio component $m(x_{i\tau})$. We assume invariance $c(m, m') = 1$ and always sample two such transformations $m_v$ and $m_a$ to extract both modalities, so $K_m = 2$. The fourth and final component $g$ applies a spatial and aural **augmentation** $TD = g(m(x_{i\tau}))$, also normalizing the data. We assume invariance $c(g, g') = 1$ and pick $K_g = 1$. The transformation $g$ comprises a pair of augmentations $(g_v, g_a)$, where $g_v(v)$ extracts a fixed-size tensor by resizing to a fixed resolution a random spatial crop of the input video $v$, and $g_a(a)$ extracts a spectrogram representation of the audio signal followed by SpecAugment (Park et al., 2019) with frequency and time masking. These choices lead to $K = K_i K_\tau K_m K_g = 4K_i$ transformations $T$ in the batch $\mathcal{T}$.

**Testing invariance and distinctiveness hypotheses.** The transformations given above combine cues that were partly explored in prior work, contrastive and non-contrastive. For example, Korbar et al. (2018) (not noise-contrastive) learns to detect temporal shifts across modalities. With our formulation, we can test whether distinctiveness or invariance to shifts is preferable, simply by setting $c(\tau, \tau') = 1$ or $c(\tau, \tau') = \delta_{\tau=\tau'}$ (this is illustrated in fig. 1). We can also set $w(\tau, \tau') = 0$ for $\tau \neq \tau'$ to ignore comparisons that involve different temporal shifts. We also test distinctiveness and invariance to **time reversal** (Wei et al., 2018), which has not previously been explored cross-modally, or contrastively. This is given by a transformation $r \in \mathcal{R} = \{r_0, r_1\}$, where $r_0$ is the identity and $r_1$ flips the time dimension of its input tensor. We chose these transformations, time reversal and time shift, because videos, unlike images, have a temporal dimension and we hypothesize that these signals are very discriminative for representation learning.

**Ignoring comparisons.** Another degree of freedom is the choice of weighting function $w(T, T')$. Empirically, we found that cross-modal supervision is a much stronger signal than within-modality supervision, so if $T$ and $T'$ slice the same modality, we set $w(T, T') = 0$ (see Appendix for ablation).

**Understanding combinations.** Finally, one may ask what is the effect of combining several different transformations in learning the representation $f$. A first answer is the rule given in eq. (2) to combine individual contrasts $c(t_m, t'_m)$ in a consistent manner. Because of this rule, to a first approximation, $f$ possesses the union of the invariances and distinctivenesses of the individual factors. To obtain a more accurate answer, however, one should also account for the details of the batch sampling scheme and of the choice of weighing function $w$. This can be done by consulting the diagrams given in fig. 1 by: (1) choosing a pair of transformations $T_i$ and $T_j$, (2) checking the value in the table (where 1 stands for invariance, 0 for distinctiveness and $\cdot$ for ignoring), and (3) looking up the composition of $T_i$ and $T_j$ in the tree to find out the sub-transformations that differ between them as the source of invariance/distinctiveness.

## 4 EXPERIMENTS

We compare self-supervised methods on pretraining audio-visual representations. Quality is assessed based on how well the pretrained representation transfers to other (supervised) downstream tasks. We first study the model in order to determine the best learning transformations and setup. Then, we use the latter to train for longer and compare them to the state of the art.

**Self-supervised pretraining.** For pretraining, we consider the standard audio-visual pretraining datasets, Kinetics-400 (Kay et al., 2017) and AudioSet (Gemmeke et al., 2017), and additionally, the recently released, VGG-Sound dataset (Chen et al., 2020a). Finally, we also explore how our algorithm scales to even larger, less-curated datasets and train on IG65M (Ghadiyaram et al., 2019) as done in XDC (Alwassel et al., 2020).

Our method learns a pair of representations $f = (f_v, f_a)$ for visual and audio information respectively and we refer to Appendix A.6 for architectural details.

**Downstream tasks.** To assess the **visual** representation $f_v$, we consider standard action recognition benchmark datasets, UCF-101 (Soomro et al., 2012) and HMDB-51 (Kuehne et al., 2011b). We test the performance of our pretrained models on the tasks of finetuning the pretrained representation, conducting few-shot learning and video action retrieval. To assess the **audio** representation $f_a$, we train a linear classifier on frozen features for the common ESC-50 (Piczak, 2015) and DCASE2014 (Stowell et al., 2015) benchmarks and finetune for VGG-Sound (Chen et al., 2020a). The full details are given in the Appendix.

### 4.1 ANALYSIS
#### OF GENERALIZED TRANSFORMATIONS

In this section, we conduct an extensive study on each parameter of the GDT transformation studied here, $T = (i, \tau, m, g)$, and evaluate the performance by finetuning our network on the UCF-101 and HMDB-51 action recognition benchmarks.

**Sample distinctiveness and invariances.** First, we experiment with extending SimCLR to video data, as shown in Table 1(a)-(d). This is an important base case as it is the standard approach followed by all recent self-supervised methods (Chen et al., 2020b; He et al., 2019; Wu et al., 2018).

Table 1: **Learning hypothesis ablation.** Results on action classification performance on HMDB-51 is shown for finetuning accuracy (Acc) and frozen action retrieval (recall@5). GDT can leverage signals from both invariance and stronger variance transformation signals, that sole data-sample (DS) variance misses.

| | DS | TR | TS | Mod. | **Acc** | **r@5** |
|---|---|---|---|---|---|---|
| *SimCLR: DS-variance only* | | | | | | |
| (a) | **v** | · | · | V | 47.1 | 32.5 |
| (b) | **v** | **i** | · | V | 39.5 | 31.9 |
| (c) | **v** | · | **i** | V | 46.9 | 34.5 |
| (d) | **v** | **i** | **i** | V | 46.6 | 33.4 |
| *GDT: 1-variance* | | | | | | |
| (e) | **v** | · | · | AV | 56.9 | 49.3 |
| (f) | **v** | **i** | · | AV | 56.1 | 49.7 |
| (g) | **v** | · | **i** | AV | 57.2 | 45.2 |
| (h) | **v** | **i** | **i** | AV | 56.6 | 44.8 |
| *GDT: 2-variances* | | | | | | |
| (i) | **v** | **i** | **v** | AV | 57.5 | 46.8 |
| (j) | **v** | **v** | **i** | AV | 57.0 | 46.2 |
| (k) | **v** | · | **v** | AV | 58.0 | 50.2 |
| (l) | **v** | **v** | · | AV | 58.2 | 50.2 |
| *GDT: 3-variances* | | | | | | |
| (m) | **v** | **v** | **v** | AV | 60.0 | 47.8 |

For this, consider GDT of the type $T = (i, m, \tau, g)$ described above and set $K_i = 768$ (the largest we can fit in our setup), $K_m = 1$ (only visual modality) and $K_g = 1$ and only pick a single time shift $K_\tau = 1$. We also set all transformation components to invariance ($c(t_m, t'_m) = 1$) except the first that does sample selection. Comparing row (a) to (b-d), we find that adding invariances to time-shift (TS) and time-reversal (TR) consistently degrades the performance compared to the baseline in (a).

**GDT variances and invariances** Our framework allows fine-grained and expressive control of which invariance and distinctiveness are learned. To demonstrate this flexibility, we first experiment with having a single audio-visual (AV) invariance transformation, in this case data-sampling (DS), i.e. $T = (i, \tau, m, g)$. We find immediately an improvement in finetuning and retrieval performance compared to the SimCLR baselines, due to the added audio-visual invariance. Second, we also find that adding invariances to TR and TS does not yield consistent benefits, showing that invariance to these transformations is not a useful signal for learning.

In rows (i-l), we explore the effect of being variant to two transformations, which is unique to our method. We find that: (1) explicitly encoding variance improves representation performance for the TS and TR transformations (58.0 and 58.2 vs 56.9). (2) Ignoring (·) the other transformation as

Table 2: **Retrieval and Few Shot Learning.** Retrieval accuracy in (%) via nearest neighbors and few shot learning accuracy (%) via training a linear SVM on fixed representations.

|  | HMDB | | UCF | |
|---|---|---|---|---|
|  | 1 | 20 | 1 | 20 |
| *Few-shot* Random | 3.0 | 4.5 | 2.3 | 6.8 |
| 3DRot (Jing & Tian, 2018) | – | – | 15.0 | 47.1 |
| **GDT (ours)** | **13.4** | **20.8** | **26.3** | **49.4** |
| *Retrieval* ClipOrder (Xu et al., 2019) | 7.6 | 48.8 | 14.1 | 51.1 |
| VCP (Cho et al., 2020) | 7.6 | 53.6 | 18.6 | 53.5 |
| **GDT (ours)** | **25.4** | **75.0** | **57.4** | **88.1** |

Table 3: **Audio classification.** Downstream task accuracies on standard audio classification benchmarks.

| Method | Acc% | |
|---|---|---|
|  | DC | ESC |
| ConvRBM (Sailor et al., 2017) | - | 86.5 |
| AVTS (Korbar et al., 2018) | 94 | 82.3 |
| DMC (Hu et al., 2019) | – | 82.6 |
| XDC Alwassel et al. (2020) | 95 | 84.8 |
| AVID (Morgado et al., 2020) | 96 | **89.2** |
| **GDT** (ours) | **98** | 88.5 |
| Human (Piczak, 2015) | – | 81.3 |

opposed to forcefully being invariant to it works better (58.2 vs 57.0 and 58.0 vs 57.5). Finally, row (m), shows the (DS, TR, TS)-variance case, yields the best performance when finetuned and improves upon the initial SimCLR baseline by more than 12% in accuracy and more than 15% in retrieval @5 performance. (DS, TR, TS) Compared to row (l), we find that using three variances compared to two does give boost in finetuning performance (58.2 vs 60.0), but there is a slight decrease in retrieval performance (50.2 vs 47.8). We hypothesize that this decrease in retrieval might be due to the 3-variance model becoming more tailored to the pretraining dataset and, while still generalizeable (which the finetuning evaluation tests), its frozen features have a slightly higher domain gap compared to the downstream dataset.

**Intuition** While we only analyse a subset of possible transformations for video data, we nevertheless find consistent signals: While both time-reversal and time-shift could function as a meaningful invariance transformation to provide the model with more difficult positives *a-priori*, we find that using them instead to force *variances* consistently works better. One explanation for this might be that there is useful signal in being distinct to these transformations. E.g., for time-reversal, opening a door carries different semantics from from closing one, and for time-shift, the model might profit from being able to differentiate between an athlete running vs an athlete landing in a sandpit, which could be both in the same video. These findings are noteworthy, as they contradict results from the image self-supervised learning domain, where learning pretext-invariance can lead to more transferable representations (Misra & van der Maaten, 2020). This is likely due to the fact that time shift and reversal are useful signals that both require learning strong video representations to pick up on. If instead invariance is learned against these, the "free" information that we have from construction is discarded and performance degrades. Instead, GDT allows one to leverage these strong signals for learning robust representations.

### 4.2 COMPARISON TO THE STATE OF THE ART

Given one of our best learning setups from Sec. 4.1 (row (l)), we train for longer and compare our feature representations to the state of the art in common visual and aural downstream benchmarks.

**Downstream visual benchmarks.**
For **video retrieval** we report recall at 1, 5, 20 retrieved samples for split-1 of the HMDB-51 and UCF-101 datasets in table 2 (the results for recall at 10 and 50 are provided in the Appendix). Using our model trained on Kinetics-400, GDTsignificantly beats all other self-supervised methods by a margin of over 35% for both datasets.

For **few-shot classification**, as shown in table 2, we significantly beat the RotNet3D baseline on UCF-101 by more than 10% on average for each shot with our Kinetics-400 pretrained model.

For **video action recognition**, we finetune our GDT pretrained network for UCF-101 and HMDB-51 video classification, and compare against state-of-the-art self-supervised methods in table 4. When constrained to pretraining on the Kinetics datasets, we find that our GDT pretrained model achieves very good results, similar to Morgado et al. (2020) (developed concurrently to our own work). When

Table 4: **State-of-the-art on video action recognition.** Self- and fully-supervisedly trained methods on UCF-101 and HMDB-51 benchmarks. We follow the standard protocol and report the average top-1 accuracy over the official splits for finetuning the whole network. Methods with [†]: use video titles as supervision, with [*]: use ASR generated text. See table A.3 for an extended version including recent/concurrent works.

| Method | Architecture | Pretraining | Top-1 Acc% | |
| --- | --- | --- | --- | --- |
| | | | HMDB | UCF |
| Full supervision (Alwassel et al., 2020) | R(2+1)D-18 | Kinetics-400 | 65.1 | 94.2 |
| Full supervision (ours) | R(2+1)D-18 | Kinetics-400 | 70.4 | 95.0 |
| *Using Kinetics* | | | | |
| AoT (Wei et al., 2018) | T-CAM | Kinetics-400 | - | 79.4 |
| XDC (Alwassel et al., 2020) | R(2+1)D-18 | Kinetics-400 | 52.6 | 86.8 |
| AV Sync+RotNet (Xiao et al., 2020) | AVSlowFast | Kinetics-400 | 54.6 | 87.0 |
| AVTS (Korbar et al., 2018) | MC3-18 | Kinetics-400 | 56.9 | 85.8 |
| CPD (Li & Wang, 2020)[†*] | 3D-Resnet50 | Kinetics-400 | 57.7 | 88.7 |
| AVID (Morgado et al., 2020) | R(2+1)D-18 | Kinetics-400 | **60.8** | 87.5 |
| **GDT (ours)** | R(2+1)D-18 | Kinetics-400 | 60.0 | **89.3** |
| *Using other datasets* | | | | |
| MIL-NCE (Miech et al., 2020)* | S3D | HowTo100M | 61.0 | 91.3 |
| AVTS (Korbar et al., 2018) | MC3-18 | AudioSet | 61.6 | 89.0 |
| XDC (Alwassel et al., 2020) | R(2+1)D-18 | AudioSet | 63.7 | 93.0 |
| AVID (Morgado et al., 2020) | R(2+1)D-18 | AudioSet | 64.7 | 91.5 |
| ELo (Piergiovanni et al., 2020) | R(2+1)D-50x3 | Youtube-2M | 67.4 | 93.8 |
| XDC (Alwassel et al., 2020) | R(2+1)D-18 | IG65M | 68.9 | **95.5** |
| **GDT (ours)** | R(2+1)D-18 | VGGSound | 61.9 | 89.4 |
| **GDT (ours)** | R(2+1)D-18 | AudioSet | **66.1** | 92.5 |
| **GDT (ours)** | R(2+1)D-18 | IG65M | **72.8** | 95.2 |

constrained to pretraining on the AudioSet (Gemmeke et al., 2017) dataset, we also find state-of-the-art performance among all self-supervised methods, particularly on HMDB-51.

We get similar performance to XDC on UCF-101. Lastly, we show the scalability and flexibility of our GDT framework by pretraining on the IG65M dataset (Ghadiyaram et al., 2019). With this, our visual feature representation sets a new state of the art among all self-supervised methods, particularly by a margin of $> 4\%$ on the HMDB-51 dataset. On UCF-101, we set similar state-of-the-art performance with XDC. Along with XDC, we beat the Kinetics supervised pretraining baseline using the same architecture and finetuning protocol.

For **audio classification** we find that we achieve state-of-the-art performance among all self-supervised methods on both DCASE2014 (DC) and ESC-50 (ESC), and also surpass supervised performance on VGG-Sound with $54.8\%$ mAP and $97.5\%$ AUC (see Tab. 5).

Table 5: **VGG-Sound.** Audio classification metrics after full-finetuning.

| Method | mAP | AUC | d' |
| --- | --- | --- | --- |
| Supervised | 51.6 | 96.8 | 2.63 |
| **GDT** (ours) | **54.8** | **97.5** | **2.77** |

## 5 CONCLUSION

We introduced the framework of Generalized Data Transformations (GDTs), which allows one to capture, in a single noise-contrastive objective, cues used in several prior contrastive and non-contrastive learning formulations, as well as easily incorporate new ones. The framework shows how new meaningful combinations of transformations can be obtained, encoding valuable invariance and distinctiveness that we want our representations to learn. Following this methodology, we achieved state-of-the-art results for self-supervised pretraining on standard downstream video action recognition benchmarks, even surpassing supervised pretraining. Overall, our method significantly increases the expressiveness of contrastive learning for self-supervision, making it a flexible tool for many multi-modal settings, where a large pool of transformations exist and an optimal combination is sought.

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

## A  APPENDIX

### A.1  THEORY

Full knowledge of the contrast function $c$ only specifies the level sets of the representation $f$.

**Lemma 1.** *The contrast $c(x_1, x_2) = \delta_{f(x_1)=f(x_2)}$ defines $f = \iota \circ \hat{f}$ up to an injection $\iota : \mathcal{X}/f \to \mathcal{Y}$, where $\mathcal{X}/f$ is the quotient space and $\hat{f} : \mathcal{X} \to \mathcal{X}/f$ is the projection on the quotient.*

*Proof.* This is a well known fact in elementary algebra. Recall that the quotient $\mathcal{X}/f$ is just the collection of subsets $X \subset \mathcal{X}$ where $f(x)$ is constant. It is easy to see that this is a partition of $\mathcal{X}$. Hence, we can define the map $\hat{f} : X \mapsto f(x)$ where $x$ is any element of $X$ (this is consistent since $f(x)$ has, by definition, only one value over $X$). Furthermore, if $\iota : x \mapsto X = \{x \in \mathcal{X} : f(x') = f(x)\}$ is the projection of $x$ to its equivalence class $X$, we have $f(x) = \hat{f}(\iota(x))$. $\qquad\square$

**Lemma 2.** *$c(x_1, x_2) = 1$ is an equivalence relation if, and only if, there exists a function $f$ such that $c(x_1, x_2) = \delta_{f(x_1)=f(x_2)}$.*

*Proof.* If $c(x_1, x_2) = 1$ defines an equivalence relation on $\mathcal{X}$, then such a function is given by the projection on the quotient $\hat{f} : \mathcal{X} \to \mathcal{X}/c = \mathcal{Y}$. On the other hand, setting $c(x_1, x_2) = \delta_{f(x_1)=f(x_2)} = 1$ for any given function $f$ is obviously reflexive, symmetric and transitive because the equality $f(x_1) = f(x_2)$ is. $\qquad\square$

The following lemma suggests that defining a contrast $c(T, T')$ on transformations instead of data samples is usually acceptable.

**Lemma 3.** *If $c(T, T') = 1$ defines an equivalence relation on GDTs, and if $TD = TD' \Rightarrow T = T'$ (i.e. different transformations output different samples), then setting $c(TD, T'D) = c(T, T')$ defines part of an admissible sample contrast function.*

*Proof.* If $x = TD$, $x' = T'D$ are obtained from some transformations $T$ and $T'$, then these must be unique by assumption. Thus, setting $c(x, x') = c(T, T')$ is well posed. Reflectivity, symmetry and transitivity are then inherited from the latter. $\qquad\square$

**Lemma 4.** *Let $c(t_m, t'_m) = 1$ be reflexive, symmetric and transitive. Their product $c(T, T') = \prod_{m=1}^{M} c(t_m, t'_m) =$ has then the same properties.*

*Proof.* The reflexive and symmetric properties are obviously inherited. For the transitive property, note that $c(T, T') = 1$ if, and only if, $\forall m : c(t_m, t'_m) = 1$. Then consider:

$$c(T, T') = c(T', T'') = 1 \quad \Rightarrow \quad \forall m : c(t_m, t'_m) = c(t'_m, t''_m) = 1$$
$$\Rightarrow \quad \forall m : c(t_m, t''_m) = 1 \quad \Rightarrow \quad c(T, T'') = 1.$$

$\square$

### A.2  GENERALITY OF GDT

Here, we show that our GDT formulation can encapsulate and unify other self-supervised works in the literature. We break it down it into two sections:

**Mapping contrastive to GDT contrastive**  Recently, a number of papers have presented contrastive formulations for image representation learning such as, NPID (Wu et al., 2018), PIRL (Misra & van der Maaten, 2020), MoCo (He et al., 2019) and SimCLR (Chen et al., 2020b). These methods are all essentially built on what we have introduced as the "data-sampling transformation" $\mathcal{T} = (i, g)$, that samples an image with index $i$ and applies augmentation $g$. For NPID, MoCo and SimCLR, the main objective is to solely be distinctive to the image index, hence $K = K_i K_g = B$ (i.e. the batchsize $B$) for NPID, due to the use of a memorybank and $K = K_i K_g = 2B$ for SimCLR and MoCo. For PIRL, one additional transformation to be invariant to is added. For example, in the case of rotation, the PIRL encodes sample-distinctiveness to the non-rotated inputs

$K = K_i K_g = B$ in the memorybank, while the rotated examples are used for constructing both invariance to the original inputs, as well as sample distinctiveness.

**Non-contrastive to GDT contrastive reduction.** In non-contrastive self-supervised formulations, one trains $\Phi(x) = y$ to regress $y$ from $x$, where $y$ is some "pretext" task label. These labels can be obtained from the data, e.g. arrow of time (Wei et al., 2018), rotation (Gidaris et al., 2018; Jing & Tian, 2018), shuffled frames (Misra et al., 2016), jigsaw configurations (Kim et al., 2019; Noroozi et al., 2017), or playback speed (Benaim et al., 2020; Cho et al., 2020).

We can reduce these pretext tasks to GDTs in two ways. The first 'trivial' reduction amounts to interpreting the supervision $y$ as an additional pseudo-modality. Consider for example RotNet; in this case, the label $y$ should record the amount of rotation applied to the input image. We can achieve this effect by starting from data $z = (x, 0)$ where $x$ is an image and $0$ a rotation angle. We then sample transformation $t_r$ (rotation) and define its action as $t_r(z) = (t_r(x), t_r(0))$ where $t_r(0) = r$ is simply the rotation angle applied and $t_r(x)$ the rotated image. We consider modality slicing transformations $m_x(z) = x$ and $m_r(z) = r$. To form a batch, we sample GDTs of the type $T = (i, t_r, m)$, where $i$ is sampled at random, for each $i$, $t_r$ is exhaustively sampled in a set of four rotations (0, 90, 180, 270 degrees) and, for each rotation $t_r$, $m$ is also exhaustively sampled, for a total of $K_i K_r K_m = 8K_i$ transformations in the batch. We define $c(T, T') = c((i, t_r, m), (i', t_{r'}, m')) = \delta_{r=r'}$ (note that we do *not* learn to distinguish different images; GDTs allow us to express this case naturally as well). We define $w(T, T') = \delta_{i=i'} \delta_{m \neq m'}$ so that images are treated independently in the loss and we always compare a pseudo modality (rotated image) with the other (label). Finally, the network $f_r(r) = e_r \in \{0, 1\}^4$ operating on the label pseudo-modality trivially encodes the latter as a 1-hot vector. Then we see that the noise-contrastive loss reduces to

$$\sum_i \sum_r \log \frac{\exp\langle f(t_r(x_i)), e_r \rangle}{\sum_{r'} \exp\langle f(t_r(x_i)), e_{r'} \rangle} \tag{3}$$

which is nearly exactly the same as a softmax loss for predicting the rotation class applied to an image.

There are other reductions as well, which capture the spirit if not the letter of a training signal. For instance, in RotNet, we may ask if two images are *rotated by the same amount*. This is an interesting example as we do *not* wish to be distinctive to which image sample is taken, only to which rotation is applied. This can also be captured as a GDT because the sampling process itself is a transformation. In this case, the set of negatives will be the images rotated by a different amount, while the positive example will be an image rotated by the same amount.

Thus, pretext task-originating transformations that have not even been explored yet can be put into our framework and, as we show in this paper, be naturally combined with other transformations leading to even stronger representations.

### A.2.1 POTENTIAL APPLICATION TO TEXT-VIDEO LEARNING

While we focus on audio-visual representation learning due to the multitude of potentially interesting learning signals, it is also possible to apply our framework to other multi-modal settings, such as video-text. Instead of a ResNet-9 as audio encoder, a text-encoder such as word-embeddings (Mikolov et al., 2013; Pennington et al., 2014) with an MLP or a transformer (Vaswani et al., 2017) can be used for encoding the textual inputs and we can train with a cross-modal NCE loss as done currently for audio-visual representation learning in our GDT framework. While the visual transformations can be kept as described in the paper, we can use transformations for text, such as sentence shuffling (Wei & Zou, 2019), or random word swaps (Wei & Zou, 2019). Moreover, unlike prior works in the literature (Alayrac et al., 2020; Li & Wang, 2020; Miech et al., 2019), which mostly focused on model and loss improvements for video-text learning, our framework would allow us to investigate whether it is more desirable to encode either invariance or disctinctiveness to these text transformations for effective video-text representation learning.

### A.3 MODALITY ABLATION

In Table A.1, we provide the results of running our baseline model (sample-distinctiveness only) within-modally instead of across modalities and find a sharp drop in performance.

Table A.1: **Multi-modal learning**, $m_m$.

| Modalities | HMDB | | UCF | |
|---|---|---|---|---|
| Epochs | 50 | 100 | 50 | 100 |
| Within-modal | 29.1 | 32.9 | 68.3 | 72.2 |
| Cross-modal | 55.1 | 56.9 | 85.1 | 87.9 |

## A.4 DATASET DETAILS

The Kinetics-400 dataset (Kay et al., 2017) is human action video dataset, consisting of 240k training videos, with each video representing one of 400 action classes. After filtering out videos without audio, we are left with 230k training videos, which we use for pretraining our model.

VGGSound (Chen et al., 2020a) is a recently released audio-visual dataset consisting of 200k short video clips of audio sounds, extracted from videos uploaded to YouTube. We use the training split after filtering out audio (170k) for pretraining our model.

Audioset (Gemmeke et al., 2017) is a large-scale audio-visual dataset of 2.1M videos spanning 632 audio event classes. We use the training split (1.8M) for pretraining our model.

IG65M (Ghadiyaram et al., 2019) is a large-scale weakly supervised dataset collected from a social media website, consisting of 65M videos of human action events. We use the all the videos in the dataset for pretraining.

HMDB-51 (Kuehne et al., 2011a) consists of 7K video clips spanning 51 different human activities. HMDB-51 has three train/test splits of size 5k/2k respectively.

UCF-101 (Soomro et al., 2012) contains 13K videos from 101 human action classes, and has three train/test splits of size 11k/2k respectively.

ESC-50 (Piczak, 2015) is an environmental sound classification dataset which has 2K sound clips of 50 different audio classes. ESC-50 has 5 train/test splits of size 1.6k/400 respectively.

DCASE2014 (Stowell et al., 2015) is an acoustic scenes and event classification dataset which has 100 training and 100 testing sound clips spanning 10 different audio classes.

## A.5 PREPROCESSING DETAILS

The video inputs are 30 consecutive frames from a randomly chosen starting point in the video. These frames are resized such that the shorter side is between 128 and 160, and a center crop of size 112 is extracted, with no color-jittering applied. A random horizontal flip is then applied with probability 0.5, and then the inputs' channels are z-normalized using mean and standard deviation statistics calculated across each dataset.

One second of audio is processed as a $1 \times 257 \times 99$ image, by taking the log-mel bank features with 257 filters and 199 time-frames after random volume jittering between 90% and 110% is applied to raw waveform, similar to (Arandjelovic & Zisserman, 2017). The spectrogram is then Z-normalized, as in (Korbar et al., 2018). Spec-Augment is then used to apply random frequency masking to the spectrogram with maximal blocking width 3 and sampled 1 times. Similarly, time-masking is applied with maximum width 6 and sampled 1 times.

## A.6 PRETRAINING DETAILS

We use R(2+1)D-18 (Tran et al., 2018) as the visual encoder $f_v$ and ResNet (He et al., 2016) with 9 layers as the audio encoder $f_a$ unless otherwise noted; both encoders produce a fixed-dimensional output (512-D) after global spatio-temporal average pooling. Both vectors are then passed through two fully-connected layers with intermediate size of 512 to produce 256-D embeddings as in (Bachman et al., 2019) which are normalized by their L2-norm (Wu et al., 2018). The embedding is used for computing the contrastive loss, while for downstream tasks, a linear layer after the global spatio-temporal average pooling is randomly intialized. For NCE contrastive learning, the temperature $\rho$ is set as $1/0.07$. For optimizing these networks, we use SGD. The SGD weight decay is $10^{-5}$ and

the SGD momentum is 0.9. We use a mini-batch size of 12 on each of our 64 GPUs giving an effective batch size of 768 for distributed training. The initial learning rate is set to 0.01 which we linearly scale with the number of GPUs, after following a gradual warm-up schedule for the first 10 epochs (Goyal et al., 2017). For both Kinetics and VGG-Sound, we train for 200 epochs (3 days), while for Audioset and IG65M, we train for 50 epochs (5 days) and 2 epochs (7 days) respectively.

## A.7 ABLATION EXPERIMENT DETAILS

For the ablations, we only train for 100 epochs on the Kinetics-400 dataset.

For both downstream tasks, we only evaluate on the first fold each but found the performance between folds to be close (within 1-2%).

## A.8 FULL VIDEO ACTION RETRIEVAL TABLE

In Table A.2 we show the full table on video action retrieval and compare to several of our models, pretrained on different datasets.

Table A.2: **Full retrieval table.**

| | HMDB | | | | | UCF | | | | |
|---|---|---|---|---|---|---|---|---|---|---|
| Recall @ | 1 | 5 | 10 | 20 | 50 | 1 | 5 | 10 | 20 | 50 |
| Supervised (Kinetics) | 49.1 | 74.4 | 83.9 | 90.6 | 96.4 | 86.9 | 94.6 | 96.5 | 98.1 | 99.0 |
| ST-Puzzle (Kim et al., 2019) | – | – | – | – | – | 19.7 | 28.5 | 33.5 | 40.0 | 49.4 |
| OPN (Lee et al., 2017) | – | – | – | – | – | 19.9 | 28.7 | 34.0 | 40.6 | 51.6 |
| ST Order (Buchler et al., 2018) | – | – | – | – | – | 25.7 | 36.2 | 42.2 | 49.2 | 59.5 |
| ClipOrder (Xu et al., 2019) | 7.6 | 22.9 | 34.4 | 48.8 | 68.9 | 14.1 | 30.3 | 40.4 | 51.1 | 66.5 |
| SpeedNet (Benaim et al., 2020) | – | – | – | – | – | 13.0 | 28.1 | 37.5 | 49.5 | 65.0 |
| VCP (Luo et al., 2020) | 7.6 | 24.4 | 36.3 | 53.6 | 76.4 | 18.6 | 33.6 | 42.5 | 53.5 | 68.1 |
| VSP (Cho et al., 2020) | 10.3 | 26.6 | 38.8 | 54.6 | 76.8 | 24.6 | 41.9 | 51.3 | 62.7 | 76.9 |
| **GDT (Kinetics)** | **25.4** | **51.4** | **63.9** | **75.0** | **87.8** | **57.4** | **73.4** | **80.8** | **88.1** | **92.9** |
| **GDT (VGG-Sound)** | **28.4** | **55.1** | **67.2** | **79.3** | **91.1** | **63.4** | **79.6** | **85.0** | **90.1** | **95.2** |
| **GDT (Audioset)** | **30.6** | **58.0** | **69.8** | **79.9** | **91.0** | **65.9** | **82.6** | **88.2** | **92.2** | **96.6** |
| **GDT (IG65M)** | **36.1** | **61.1** | **70.8** | **79.7** | **92.1** | **75.7** | **87.2** | **90.7** | **93.5** | **96.6** |

## A.9 FULL VIDEO ACTION RECOGNITION TABLE

Table A.3: **State-of-the-art on action recognition.** Self-supervised and supervised methods on UCF101 and HMDB51 benchmarks. We follow the standard protocol and report the average top-1 accuracy over the official splits and show results for finetuning the whole network. Note that we find the supervised baseline to be around 6% and 2% better than reported in (Alwassel et al., 2020) as we use a different finetuning strategy. Methods with [†] indicate the additional use of video titles as supervision. Methods with [*] use ASR generated text. Methods in gray are concurrent works.

| Method | Architecture | Pretrain Dataset | Top-1 Acc% HMDB | Top-1 Acc% UCF |
|---|---|---|---|---|
| Full supervision | R(2+1)D-18 | ImageNet | 46.7 | 82.8 |
| Full supervision (Alwassel et al., 2020) | R(2+1)D-18 | Kinetics-400 | 65.1 | 94.2 |
| Full supervision (ours) | R(2+1)D-18 | Kinetics-400 | 70.4 | 95.0 |
| Full supervision (Tran et al., 2018) | R(2+1)D-34 | Kinetics-400 | 74.5 | 96.8 |
| *Using UCF* | | | | |
| Shuffle and Learn (Misra et al., 2016) | CaffeNet | UCF | 18.1 | 50.2 |
| VGAN (Vondrick et al., 2016) | VGAN | Flickr | – | 52.1 |
| LT-Motion (Luo et al., 2017) | VGG-16 | UCF | – | 53.0 |
| Geometry (Gan et al., 2019) | CaffeNet | UCF | 23.3 | 55.1 |
| OPN (Lee et al., 2017) | VGG | UCF | 23.8 | 56.3 |
| ST Order (Buchler et al., 2018) | CaffeNet | UCF | 25.0 | 58.6 |
| CMC (Tian et al., 2019) | CaffeNet | UCF | 26.7 | 59.1 |
| VCP (Luo et al., 2020) | R(2+1)D-18 | UCF | 32.2 | 66.3 |
| Cross and Learn (Sayed et al., 2018) | VGG-16 | UCF | 33.0 | 70.5 |
| *Using Kinetics* | | | | |
| ClipOrder (Xu et al., 2019) | R(2+1)D-18 | Kinetics-400 | 30.9 | 72.4 |
| MotionPred (Wang et al., 2019) | C3D | Kinetics-400 | 33.4 | 61.2 |
| RotNet3D (Jing & Tian, 2018) | 3D-ResNet18 | Kinetics-600 | 33.7 | 62.9 |
| ST-Puzzle (Kim et al., 2019) | 3D-ResNet18 | Kinetics-400 | 33.7 | 65.8 |
| DPC (Han et al., 2019) | 3D-ResNet34 | Kinetics-400 | 35.7 | 75.7 |
| VPS (Cho et al., 2020) | R3D | Kinetics-400 | 36.8 | 74.8 |
| SpeedNet (Benaim et al., 2020) | I3D | Kinetics-400 | 43.7 | 66.7 |
| AoT (Wei et al., 2018) | T-CAM | Kinetics-400 | - | 79.4 |
| CBT (Sun et al., 2019a) | S3D | Kinetics-600 | 44.6 | 79.5 |
| Multisensory (Owens & Efros, 2018) | 3D-ResNet18 | Kinetics-400 | - | 82.1 |
| XDC (Alwassel et al., 2020) | R(2+1)D-18 | Kinetics-400 | 52.6 | 86.8 |
| AV Sync+RotNet (Xiao et al., 2020) | AVSlowFast | Kinetics-400 | 54.6 | 87.0 |
| AVTS (Korbar et al., 2018) | MC3-18 | Kinetics-400 | 56.9 | 85.8 |
| CPD (Li & Wang, 2020)[†*] | 3D-Resnet50 | Kinetics-400 | 57.7 | 88.7 |
| AVID (Morgado et al., 2020) | R(2+1)D-18 | Kinetics-400 | 60.8 | 87.5 |
| CoCLR (Han et al., 2020) | S3D | Kinetics-400 | **62.9** | **90.6** |
| **GDT (ours)** | R(2+1)D-18 | Kinetics-400 | 60.0 | 89.3 |
| *Using other datasets* | | | | |
| L³-Net (Arandjelovic & Zisserman, 2017) | VGG-16 | AudioSet | 40.2 | 72.3 |
| Speech2Action* (Nagrani et al., 2020) | S3D-G | MovieDataset | 58.1 | - |
| DynamoNet (Diba et al., 2019) | ResNext101 | Youtube8M | 58.6 | 87.3 |
| MIL-NCE (Miech et al., 2020)* | S3D | HowTo100M | 61.0 | 91.3 |
| AVTS (Korbar et al., 2018) | MC3-18 | AudioSet | 61.6 | 89.0 |
| XDC (Alwassel et al., 2020) | R(2+1)D-18 | AudioSet | 63.7 | **93.0** |
| AVID (Morgado et al., 2020) | R(2+1)D-18 | AudioSet | 64.7 | 91.5 |
| MMV* (Alayrac et al., 2020) | R(2+1)D-18 | Audioset | **70.1** | 91.5 |
| ELo (Piergiovanni et al., 2020) | R(2+1)D-50x3 | Youtube-2M | 67.4 | 93.8 |
| XDC (Alwassel et al., 2020) | R(2+1)D-18 | IG65M | 68.9 | **95.5** |
| MMV* (Alayrac et al., 2020) | TSM-50x2 | AudioSet+HT100M | **75.0** | 95.2 |
| **GDT (ours)** | R(2+1)D-18 | VGGSound (170K) | 61.9 | 89.4 |
| **GDT (ours)** | R(2+1)D-18 | AudioSet (1.7M) | 66.1 | 92.5 |
| **GDT (ours)** | R(2+1)D-18 | IG65M | 72.8 | 95.2 |
| **GDT (ours)** (only finetune `fc`) | R(2+1)D-18 | IG65M | 55.1 | 85.4 |

## A.10 EVALUATION DETAILS

All evaluation code is provided in the Supplementary Material.

**Video**   During training, we take 10 random clips of length 32 frames from each video. For video clip augmentations, we follow a standard protocol as in (Korbar et al., 2018). During evaluation, we uniformly sample 10 clips from each video, average softmax scores, and predict the class having the highest mean softmax score. We then measure the mean video top-1 accuracy across all videos and all official folds. During training, we use SGD with initial learning rate 0.0025, which we gradually warm up to $2 \cdot 10^{-2}$ in the first 2 epochs. The weight decay is set to $5 \cdot 10^{-3}$ and momentum to 0.9. We use a mini-batch size of 32 and train for 12 epochs with the learning rate multiplied by $5 \cdot 10^{-2}$ at 6 and 10 epochs. We compare our GDT pretrained model with both self-supervised methods, and supervised pretraining, and report average top-1 accuracies on UCF101 and HMDB-51 action recognition task across three folds in table A.3.

**Few-shot classification**   We follow the protocol in (Jing & Tian, 2018) and evaluate our our GDT pretrained network using few-shot classification on the UCF-101 dataset, and additionally on HMDB-51. We randomly sample $n$ videos per class from the train set, average the encoder's global average pooling features from ten clips per training sample and measure classification accuracy performance on the validation set using a $k$-nearest neighbor classifier, with $k$ set to 1.

**Retrieval**   We follow the standard protocol as outlined in (Xu et al., 2019). We use the split 1 of UCF101, and additionally HMDB-51. We uniformly sample 10 clips per video, and average the max-pooled features after the last residual block for each clip per video. We use these averaged features from the validation set to query the videos in the training set. The cosine distance of representations between the query clip and all clips in the training set are computed. When the class of a test clip appears in the classes of $k$ nearest training clips, it is considered to be correctly predicted. We report accuracies for $k = 1, 5, 10, 20, 50$ and compare with other self-supervised methods on UCF101 and HMDB-51 in  table A.2.

**Audio**   We extract 10 equally spaced 2-second sub-clips from each full audio sample of ESC-50 (Piczak, 2015) and 60 1-second sub-clips from each full sample of DCASE2014 (Stowell et al., 2015). We save the activations that result from the audio encoder to quickly train the linear classifiers. We use activations after the last convolutional layer of the ResNet-9 and apply a max pooling with kernelsize (1,3) and stride of (1,2) without padding to the output. For both datasets, we then optimize a L2 regularized linear layer with batch size 512 using the Adam optimizer (Kingma & Ba, 2015) with learning rate $1 \cdot 10^{-4}$, weight-decay set to $5 \cdot 10^{-4}$ and the default parameters. The classification score for each audio sample is computed by averaging the sub-clip scores in the sample, and then predicting the class with the highest score. The mean top-1 accuracy is then taken across all audio clips and averaged across all official folds. For VGG-Sound (Chen et al., 2020a), we follow their evaluation metrics but follow a much shorter training schedule as our model is pretrained. We optimize the network with batch size 128 using the Adam optimizer (Kingma & Ba, 2015) with learning rate $1 \cdot 10^{-4}$ for the pretrained backbone and $1 \cdot 10^{-3}$ for the newly randomly initialized linear layer, weight-decay set to $1 \cdot 10^{-5}$ and the default parameters. We drop the learning rate at 10 and 20 epochs and train for 30 epochs, which takes less than 10h on a single Nvidia GTX 1080 Titan GPU.

