# OpenReview forum: "Multi-modal Self-Supervision from Generalized Data Transformations"
_ICLR.cc/2021/Conference — Reject_

### Official Review · AnonReviewer1 · 2020-10-24
**Recommend accept**

**Rating:** 6
**Confidence:** 4

**Review:**

This paper presents Generalized Data Transformations (GDT), a framework that unifies different self-supervised learning methods. Using this framework for unsupervised video representations, the paper gets state-of-the-art results on downstream tasks.

### Strengths:
- Unification of different self-supervised methods into a single framework, which is clean and general. This includes two main very related ideas. First, the different methods are reframed to work in a contrastive setting. Second, in order to organize the combination of the methods, the authors propose a way to do it in a systematic way.
- Following the previous framework, the paper proposes combinations of self-supervised methods for video representations, and it shows better results than competing methods. Both the baselines and ablations are sensible.
- Overall well written and easy to follow.
- Code is provided.

### Weaknesses
- It is not clear whether the main contribution is the GDT framework or its application to video. After reading the experiments section, GDT feels more like a way of structuring the experiments, and less like a unified way of understanding self-supervised learning. This feeling is reinforced by the seemingly arbitrary task the authors select (audio-visual data in videos), given the method. While video representations is a very important topic, the method does not lead to video representations as its most direct application.
- The GDT framework consists of two ideas, and they are not properly separated in the paper (or not totally unified in a single one). On the one hand, there is the formulation of different self-supervised methods as contrastive losses. On the other hand, there is the organization of data augmentation combinations. Please note that the second is not strictly necessary for the first one to work (therefore the previous point about "structuring the experiments").
- The results lack a lot of intuition and analyses. Why some data augmentations work better than others? Why opposite augmentations (being variant _and_ invariant to time shift) are useful separately? Also, the application is specific to video, so explanations and analyses of the results on video tasks should be discussed. Why do these combinations of augmentations work for video? What information are the representations encoding?
- Other unifying frameworks have been proposed for self-supervised contrastive formulations. Specifically, [1] (which is cited in the paper but not discussed) proposes to view all of these methods as multiple views of a scene (which are the positives). What conceptual contribution does this paper add on top (or instead of) the one presented in [1]?
- While the framework is theoretically clean and general, in practice there are a lot of corner cases, and exceptions. With only two augmentations (excluding the "sampling" one), a lot of the combinations are already not possible, and the authors have to propose specific ways of combining them that are very tailored to the problem. This implies that the method is general but it has a lot of hyperparameters that need to be tuned or decided manually. Similarly, the augmentations used in the paper are not systematically selected following any rules or framework, but chosen directly by the authors.

### Additional comments and questions:
- Is the reference to SimCLR incorrect? Right now it shows Tian et al 2019, in all the cases it is cited. Tian et al 2019 is the Contrastive Multiview Coding paper, not SimCLR.
- Is the format the correct one for ICLR 2021?

### Final recommendation
Overall, I believe the strengths outweigh the weaknesses and I recommend this paper to be accepted to ICLR, but I suggest the authors address the previously mentioned points.

### References
[1] Yonglong Tian, Dilip Krishnan, and Phillip Isola. _Contrastive multiview coding_. arXiv preprint arXiv:1906.05849, 2019.

---

> ### Author Response · Authors · 2020-11-13
> **Response to Reviewer 1**
>
> We thank Reviewer 1 for their valuable comments, detailed reading and the highlighting of the unifying nature of our framework, the detailed ablations and ease to follow as strengths of our paper.
>
>
> **It is not clear whether the main contribution is the GDT framework or its application to video. After reading the experiments section, GDT feels more like a way of structuring the experiments, and less like a unified way of understanding self-supervised learning. This feeling is reinforced by the seemingly arbitrary task the authors select (audio-visual data in videos), given the method. While video representations is a very important topic, the method does not lead to video representations as its most direct application.**
> Our main contributions are indeed both a general framework *and* its detailed application to the audio-visual domain.
> The framework that we develop makes explicit that it is the choice, combinations and variance and invariance properties of transformations that are key for self-supervised learning.
> In turn, our motivation for focusing on videos lies in the vastly larger space of potential transformations compared to the image domain, due to the additional time and modality dimensions. This makes audio-vidual data a more conclusive and comprehensive test for our framework. Leveraging the GDT framework, we provide a detailed analysis and empirical results on which sets of transformations yield state of the art video representation learning performance.
>
> **The GDT framework consists of two ideas, and they are not properly separated in the paper (or not totally unified in a single one). On the one hand, there is the formulation of different self-supervised methods as contrastive losses. On the other hand, there is the organization of data augmentation combinations. Please note that the second is not strictly necessary for the first one to work (therefore the previous point about "structuring the experiments").**
> By unifying the formulation of different self-supervised methods as contrastive losses, it allows us to demonstrate that the performance of self-supervised learning depends primarily on the data transformations applied, and not necessarily the choice of loss function. As for the second point, while the organization of combinations is not critical to describe *existing* methods, it is critical to describe *new* methods that work, as well as explain why some combinations *don't* work. Without this framework, it is possible to obtain combinations that are apparently benign, but we show to be nonsensical from a learning perspective. We refer the reviewer to Appendix A.1 for this proof.

---

> > ### Author Response · Authors · 2020-11-13
> > **Response to Reviewer 1 (Continued)**
> >
> > **The results lack a lot of intuition and analyses. Why some data augmentations work better than others? Why opposite augmentations (being variant and invariant to time shift) are useful separately? Also, the application is specific to video, so explanations and analyses of the results on video tasks should be discussed. Why do these combinations of augmentations work for video? What information are the representations encoding?**
> > The focus of our work lies in formulating and applying this framework for the task of video representation learning.
> > As the space of transformations in videos is much richer, our starting hypothesis was that learning video representations that are distinctive, rather than invariant, to temporal transformations -- such as time shift and reversal -- would be more effective, compared to simply learning strong invariances, which are the defacto default in the image domain.
> > Therefore, we focus on testing the effect of different transformations on representation learning performance through a rigorous empirical study facilitated by our GDT framework. Nevertheless, we agree with the reviewer and will update our paper with some intuitions behind the transformations introduced in the paper and their possible interpretations for the video context. For example, time-reversal invariance encodes the prior knowledge that some actions are time-symmetric (e.g. jumping on a trampoline), while its distinctiveness indicates the opposite (e.g. opening vs. closing a door).
> >
> >
> > **Other unifying frameworks have been proposed for self-supervised contrastive formulations. Specifically, [1] (which is cited in the paper but not discussed) proposes to view all of these methods as multiple views of a scene (which are the positives). What conceptual contribution does this paper add on top (or instead of) the one presented in [1]?**
> > CMC [1] shows that learning representations across views (such as different color channels or indeed modalities) can be an effective strategy for learning representations. However, CMC, like most previous contrastive learning methods, focuses primarily on learning *invariances* to data augmentations such as color jittering, cropping, while only being distinctive to the data-sampling transformation. In contrast, our framework clearly distinguishes transformations as learning signals to be either variant or invariant to, which we find to be more suitable to the audio-visual setting: solely using augmentations works, but significant gains can be made by incorporating variances/distinctiveness (see Table 1).
> > We will highlight this differentiation to CMC in our related works section.
> >
> >
> > **While the framework is theoretically clean and general, in practice there are a lot of corner cases, and exceptions. With only two augmentations (excluding the "sampling" one), a lot of the combinations are already not possible, and the authors have to propose specific ways of combining them that are very tailored to the problem. This implies that the method is general but it has a lot of hyperparameters that need to be tuned or decided manually. Similarly, the augmentations used in the paper are not systematically selected following any rules or framework, but chosen directly by the authors.**
> > The cleanliness and generality go hand in hand with our framework's applicability: without our hierarchical framework, many possible but non-sensical transformations can be constructed. It is our contribution to show, with a set of exemplary and common transformations, the value in using our framework to construct valid pairs of contrastive comparisons. As for deciding the combinations and hyper-parameters, while this is computationally expensive in its current form (with exhaustive search), we lay the groundwork for future research to try to perform this search more efficiently.
> >
> > **Is the reference to SimCLR incorrect? Right now it shows Tian et al 2019, in all the cases it is cited. Tian et al 2019 is the Contrastive Multiview Coding paper, not SimCLR. Is the format the correct one for ICLR 2021?**
> > Thank you for pointing this out.  We will fix this and update the manuscript accordingly.

---

### Official Review · AnonReviewer4 · 2020-10-28
**A well executed paper for audio-video self supervised learning but a potential lack of novelty**

**Rating:** 7
**Confidence:** 4

**Review:**

## Summary

The paper introduces a general framework dubbed Generalized Data Transformations (GDT) for self supervised learning. The framework is used to perform video-audio self supervised learning and analyze what kind of transformations the representations should be invariant to or on the contrary variant to thanks to a contrastive loss. The author demonstrate the effectiveness of the proposed approach by showing that the resulting learned video representations achieve very good performance on the HMDB51 and UCF101 downstream task.

## Strengths

- Overall the paper is well written
- There are some interesting findings about the paper. I am notably thinking about the results in Table 1 indicating that it is beneficial to be variant to time reversal which demonstrate that some augmentation should actually be used as negatives rather than positives in contrastive learning.
- The final results are really good

## Weaknesses

**About the GDT formulation**: The idea of trying to have a general framework that can encompass all self supervised contrastive methods is a valuable effort. However, one feeling that I have about the GDT framework is that it brings more complexity (many notations are introduced e.g. $c$ the weights $w$, the different transformations $T$,...) than it actually brings new insights and benefits.

-  I have notably the feeling that one could have written a paper that would have put more emphasis on the interesting findings of the specific multi-modal case that is explore here (video-audio) rather than trying to fit the findings into cumbersome notations. Things might have been different if more than just the setup of video-audio had been explored to better illustrate the versatility of the proposed framework.

- Also I am questioning the generality of the framework. In particular for the multimodal case, I am unsure that the actual mathematical formulation works as plugging $f=(f_v, f_a)$ in equation (1) does not work and I guess does not correspond to the actual thing that is done in the experiments. What is actually done is that $f$ changes depending on the transformation (it becomes $f_a$ if the transformation corresponds to extracting the audio and $f_v$ otherwise), but this does not seem to be completely covered by the formulation. A similar issue would arise if we wanted to have different networks for different transformation of the same modality.

In short: what are the advantages of having this framework? Did this framework helped the authors to construct new intuitions? Since it seems to be one of the main contribution of the paper this is important that the authors address that point.

**About originality**:  If we put the introduction of the GDT aside (given the previous raised point), the paper does not bring impressive conceptual innovations for training multimodal representations as the method is similar to Korbar et al. 2018, Arandjlevovic 2017 and more recently from AVID and XDC that also learn representation by using the self supervision contained in the cross modality of video and audio. In particular the loss is not novel, the architecture used to merge the modalities are not novel and the overall conclusion is in line with previous work (that the best thing seems to be to use the other modality as an extra view for learning good representations).

**About TR and TS invariances**

- If I understand correctly there is a single negative coming from the same video that has been time reversed (TR) (or time shifted TS), however there would be many more negatives coming from other videos (in the denominator of equation (1)). I wonder if it would be beneficial to try to upweight these single negatives coming from those specific transformations? Is this something that the authors have considered?

- A related question is whether or not TR and TS can be combined to obtain 3 invariances? Would that be beneficial? From an intuitive point of view it seems that the two signals could be complementary.


**Resolution of the video 112x112**

In the appendix it is mentioned that the resolution used are 112x112. What would happen if you were to use higher input resolution (e.g. 224x224). In particular in XDC this is the resolution used and this might lead to improvements in your case as well that could further improve the performance of the method.

**Linear evaluation on UCF/HMDB**: It would be nice to also evaluate the representations on the frozen setting on UCF and HMDB (as more recent methods like ELo and MILNCE are doing). This would make the comparison in Table 2 a bit more stronger than using the retrieval or the few-shot setup that was used by methods that were not leveraging multiple modalities for learning.

**Would the final trained models be available?** In particular the IG65M dataset is not open sourced so its important that the authors release the weights of the trained models.

## Conclusion and assessment

Overall the paper is well written and well executed. The results are strong for self supervised learning from audio and video. Nonetheless I have some global concerns about the work, notably the limited usefulness of the introduced general framework (GDT), and the overall lack of novel concepts or new insights provided by the work (despite the TR and TS findings that seem new to me). That is why as of now I feel the paper is borderline. I am still leaning towards acceptance since I feel the paper is an important milestone for self supervised learning from video and audio but I will wait for the answers of the authors to take a final informed decision.

## Post Rebuttal comment

The authors have clarified the contributions of their work and improved the manuscript accordingly. Given this and the other positive points about the paper I am willing to increase my score to accept.

---

> ### Author Response · Authors · 2020-11-13
> **Response to Reviewer 4**
>
> We thank the reviewer for their detailed and constructive comments and their positive feedback on our novel findings, strong performance and execution of the method.
>
> **About the GDT formulation: The idea of trying to have a general framework that can encompass all self supervised contrastive methods is a valuable effort. However, one feeling that I have about the GDT framework is that it brings more complexity (many notations are introduced e.g. the weights , the different transformations T...) than it actually brings new insights and benefits.**
> While the framework may feel unfamiliar at first, we believe that this is because it does capture the valuable intuition that most "ingredients" in current self-supervised learning formulations can be reduced to data transformations.
> Because of this intuition, we can treat all such transformations in a uniform manner based on a few simple rules (consistency constraints). In turn, this allows to search systematically for effective training signals that combine invariances and variances in novel ways. We believe that this justifies the additional notation in such studies.
>
> **I have notably the feeling that one could have written a paper that would have put more emphasis on the interesting findings of the specific multi-modal case that is explore here (video-audio) rather than trying to fit the findings into cumbersome notations. Things might have been different if more than just the setup of video-audio had been explored to better illustrate the versatility of the proposed framework.**
> While our experiments do focus on the audio-visual domain, we do intend to apply our framework to other scenarios, such as cross-modal video-text learning. While video-text learning for action recognition has been explored by works such as MIL-NCE [1] and MMV [2], their focus was more in developing novel losses and architectures rather than exploring text transformations, such as sentence shuffling [5], and random word swaps [5], that can be used to encode desirable invariances or equivariances in the video representations when training cross-modally. As a follow up to GDT, we intend to investigate which text transformations in the [nlpaug](https://github.com/makcedward/nlpaug) library are best suited to improve video representation learning, fully showing the versatility and flexibility of our GDT framework.
>
>
> **In short: what are the advantages of having this framework? Did this framework helped the authors to construct new intuitions? Since it seems to be one of the main contribution of the paper this is important that the authors address that point.**
> Yes, the framework enabled new insights. The starting point of our empirical analysis was finding out if combinations of invariance and variance to different transformations would improve performance (which in fact we do show in the experiments, see Tab.1).
> However, faced with the task of combining these ingredients, it is not obvious what combinations are possible.
> For instance, we cannot look for a representation that is simultaneously invariant to time shift and variant to time reversal if the two transformations are applied independently, as this leads to a contradiction (e.g. if we apply both time shift and reversal the representation would need to be both invariant and variant at the same time, which is not possible).
> In this case, GDT indicates that there must be an ordering or precedence between choices in order to satisfy the consistency constraints of Sec. 3. For example, a possible resolution suggested by the framework is to be variant to time reversal *unless* a time shift is also applied, in which case invariance wins; the opposite resolution is also possible.
>
>
> **About originality: If we put the introduction of the GDT aside (given the previous raised point), the paper does not bring impressive conceptual innovations for training multi-modal representations as the method is similar to Korbar et al. 2018, Arandjlevovic 2017 and more recently from AVID and XDC that also learn representation by using the self supervision contained in the cross modality of video and audio. In particular the loss is not novel, the architecture used to merge the modalities are not novel and the overall conclusion is in line with previous work (that the best thing seems to be to use the other modality as an extra view for learning good representations).**
> While we do not introduce new architectures or losses, we *do* provide several valuable contributions and innovations:
> **A.** We show that just adding invariance to more transformations is not optimal; rather, being variant (distinctive) to some is better.
> **B.** We show that most self-supervised formulations reduce to a choice between invariances and variance with respect to combinations of different transformations.
> **C.** We show that not all combinations are possible and state precisely which ones are.
> **D.** We provide a "search space" for possible formulations in contrastive learning.

---

> > ### Author Response · Authors · 2020-11-13
> > **Response to Reviewer 4 (Continued)**
> >
> > **If I understand correctly there is a single negative coming from the same video that has been time reversed (TR) (or time shifted TS), however there would be many more negatives coming from other videos (in the denominator of equation (1)). I wonder if it would be beneficial to try to upweight these single negatives coming from those specific transformations? Is this something that the authors have considered?**
> > That is a great point! We also had a similar intuition but when we tried this,  we did not find it to be beneficial and for simplicity left it out. We will include these additional negative results in the appendix.
> >
> > **A related question is whether or not TR and TS can be combined to obtain 3 invariances? Would that be beneficial? From an intuitive point of view it seems that the two signals could be complementary.**
> > A model with 3 invariances, (TS,TR and TS$\cdot$TR) could be trained for example by uniformly sampling one of these three variations when computing the positive dot products.
> > However, as we see in our experiments (Table 1) incorporating more *variances*, as opposed to invariances seems to give the most consistent benefits. This does not exclude the possibility of TS,TR and TS$\cdot$TR invariant representations being useful for some other downstream tasks (that, sat, value extreme invariance), but at least in the context of video action recognition and retrieval, this does not seem to be the case.
> >
> > **In the appendix it is mentioned that the resolution used are 112x112. What would happen if you were to use higher input resolution (e.g. 224x224). In particular in XDC this is the resolution used and this might lead to improvements in your case as well that could further improve the performance of the method.**
> > We experimented with this, but with a batch-size `B`, the input tensors of size `Bx3x30x224x224` are very difficult to fit in memory, and we found that most of the performance boosts comes from longer temporal resolution (30 vs 8 frames), rather than increasing the spatial resolution. Recent works, such as CoCLR[3] and AVID CMA[4], confirm the strong effect of the temporal resolution.
> >
> > **Linear evaluation on UCF/HMDB: It would be nice to also evaluate the representations on the frozen setting on UCF and HMDB (as more recent methods like ELo and MIL-NCE are doing). This would make the comparison in Table 2 a bit more stronger than using the retrieval or the few-shot setup that was used by methods that were not leveraging multiple modalities for learning.**
> > We agree. We have started this experiment and will add linear evaluation numbers for UCF-101 and HMDB-51 to our revised paper.
> >
> > **Would the final trained models be available? In particular the IG65M dataset is not open sourced so its important that the authors release the weights of the trained models.**
> > We will release both the code and pretrained models, including IG65M-trained ones.

---

> > > ### Author Response · Authors · 2020-11-13
> > > **Response to Reviewer 4 (Continued)**
> > >
> > > **Also I am questioning the generality of the framework. In particular for the multimodal case, I am unsure that the actual mathematical formulation works as plugging $f=(f_v,f_a)$ in equation (1) does not work and I guess does not correspond to the actual thing that is done in the experiments. What is actually done is that changes depending on the transformation (it becomes if the transformation corresponds to extracting the audio and fv otherwise), but this does not seem to be completely covered by the formulation. A similar issue would arise if we wanted to have different networks for different transformation of the same modality.**
> > > The notation $f(TD)$ in Eq. (1) is a compact way of stating that either $f_v$ or $f_a$ is applied to $TD$ depending on whether the transformation $T$ extracts the visual or audio component from the data, as the reviewer guessed.
> > > A more pedantic but correct way of writing this is to set the range of the modality-extraction transformations to be the cartesian product $(V \cup \{\emptyset\}) \times (A \cup \{\emptyset\})$ of audio and visual data modalities, including the absence of data ($\emptyset$). So for example we would define $m_a(x) = (\text{audio}(x), \emptyset)$ and $m_v(x) = (\emptyset, \text{video}(x))$ where $\emptyset$ denotes an "empty" slot. Then one defines $f(a,v) = f_a(a)$ if $v = \emptyset$ and $f(a,v)=f_v(v)$ if $a = \emptyset$ (where, by construction, exactly one of $a$ and $v$ must be equal to the $\emptyset$ placeholder; so that this definition is well posed). Note that, in our formulation, further transformations are applied after extracting the modalities and these would need to be likewise represented (see Section 3.1 in our paper for examples). Adding more slots allow to differentiate and thus incorporate any number of modalities. Splitting information in this manner is necessary because different networks/transformation types are required to process different types of data (e.g. audio vs. video) after splitting.
> > > The notation $f(TD)$ compactly captures this simple idea, which is very verbose to write down explicitly -- at the same time, there is no "hidden" limitation of the framework as the contruction above can be generalized to any number of modalities by simply adding more slots/networks.
> > >
> > >
> > > **References**
> > > [1] Antoine Miech, Jean-Baptiste Alayrac, Lucas Smaira, Ivan Laptev, Josef Sivic, Andrew Zisserman. End-to-End Learning of Visual Representations from Uncurated Instructional Videos. CVPR 2020.
> > > [2] Jean-Baptiste Alayrac, Adrià Recasens, Rosalia Schneider, Relja Arandjelović, Jason Ramapuram, Jeffrey De Fauw, Lucas Smaira, Sander Dieleman, Andrew Zisserman. Self-Supervised MultiModal Versatile Networks. NeurIPS 2020.
> > > [3] Tengda Han, Weidi Xie, Andrew Zisserman. Self-supervised Co-training for Video Representation Learning. NeurIPS 2020.
> > > [4] Pedro Morgado, Nuno Vasconcelos, Ishan Misra. Audio-Visual Instance Discrimination with Cross-Modal Agreement. ArXiv 2020.
> > > [5] Jason Wei, Kai Zou. Easy Data Augmentation. EMNLP-IJCNLP 2019.

---

> > > > ### Comment · AnonReviewer4 · 2020-11-22
> > > > **Additional references**
> > > >
> > > > Thanks for clarifying these points and the overall contribution of the paper. These additional references that you gave seem to be quite relevant but some do not appear in the revised version. In particular [2,3] could be directly added to the comparison in Table 4 of the paper. At least these work (as I understand some will only appear in NeurIPS in December only though they seemed to be on arxiv for longer) could be mentioned in the related work or some discussion mentioning the extension to other modalities for the final version of the paper.

---

> > > > > ### Author Response · Authors · 2020-11-22
> > > > > **Additional References Follow-up**
> > > > >
> > > > > We are glad we were able to make our paper contribution much clearer. Yes [2,3] are both relevant related work and will both be appearing in NeurIPS in December 2020. We will definitely add these to the final version of the paper, in the related works, SOTA comparison and the extension to other modalities sections.

---

> > > ### Comment · AnonReviewer4 · 2020-11-13
> > > **Precision**
> > >
> > > About: *A related question is whether or not TR and TS can be combined to obtain 3 invariances? Would that be beneficial? From an intuitive point of view it seems that the two signals could be complementary. A model with 3 invariances, (TS,TR and TSTR) could be trained for example by uniformly sampling one of these three variations when computing the positive dot products. However, as we see in our experiments (Table 1) incorporating more variances, as opposed to invariances seems to give the most consistent benefits. This does not exclude the possibility of TS,TR and TSTR invariant representations being useful for some other downstream tasks (that, sat, value extreme invariance), but at least in the context of video action recognition and retrieval, this does not seem to be the case.*
> > >
> > > Sorry my mistake, I meant, could you impose **variance** to both TR and TS. So far it seems that only TR variance or TS variance was tried separately but the two could be combined?

---

> > > > ### Author Response · Authors · 2020-11-14
> > > > **Good point. We will run this experiment.**
> > > >
> > > > We agree, building on our findings of using one and two variances, imposing all three DS, TR and TS and variances might indeed improve performance further.
> > > > We will run this experiment and aim to add the numbers in the revised version.

---

### Official Review · AnonReviewer3 · 2020-10-29
**Lacking methodological and technical impact**

**Rating:** 4
**Confidence:** 3

**Review:**

The authors propose to integrate a few data transformations into a generalized formulation. Similarly to the motivation of previous contrastive learning, the generalized transformation are required to learn robust representation while balancing between invariance and distinctiveness. Experiments show some validations of the proposed framework on audio-visual scenarios.

+ The authors provide a good summarization of existing contrastive augmentations and data sampling into a generalized formulation.
+ Video transformations in contrastive learning has not been carefully investigated before.
+ The raised problem of balancing (or enumerating) between distinctive vs invariant transformations is underexplored and worth studying.

- While the introduced formulation is a good wrap-up of possible contrastive augmentations, it has no practical impact until the users find the best combination through a brute-force enumeration of candidate transformations.
- I believe only the formulation is general, while their method or framework would not be generalizable to other datasets / modalities / self-tasks. With different scenarios, different combinations have to be experimented one by one.
- The experiment is done on a very specific scenario: audio-visual task, from which I believe that the main contribution of this work is more of the improvement of a specific audio-visual self-sup method, rather than a generalized formulation of the transformations.
- Minor: Quite many symbols hurt the readability.

---

> ### Author Response · Authors · 2020-11-13
> **Response to Reviewer 3**
>
> We thank Reviewer 3 for their time and feedback, and appreciate the comments that the topic of variance vs. invariance is important and understudied, and that our framework is a good unification of contrastive methods. In the following, we address all concerns raised and hope to provide clarity on the nature of our contributions.
> In particular, the technical impact of our framework lies in
> **A.** providing formal theory on which transformation combinations are consistent (beginning of Sec. 3) , and how these transformations can be constructed via our hierarchical sampling framework (see Fig. 1A)
> **B.** how to leverage this framework to explore the space of possible transformations as learning signals, in particular with respect to invariances vs variances (see. Eq.2, Tab.1)
> **C.** a detailed empirical study on using GDTs to determine the optimal choice of transformations for state-of-the-art audio-visual representation learning.
>
> **While the introduced formulation is a good wrap-up of possible contrastive augmentations, it has no practical impact until the users find the best combination through a brute-force enumeration of candidate transformations.**
> It is true that finding the best combination is empirical, but our contribution still has a practical impact because GDTs allow to systematically enumerate sensible configurations to test. Without this, it is not even possible to run a search, and therefore the value of our GDT framework actually lies in the state *before* the best combination is found. Depending on available resources, the search itself could then be carried out using any of a number of techniques, such as grid search (which we chose) or more complex Bayesian optimization. Given the set of exemplary transformations under study, we were able to find the optimal combination by enumeration, and decided to present this instead of an approximate solution with other search techniques.
>
> **I believe only the formulation is general, while their method or framework would not be generalizable to other datasets / modalities / self-tasks. With different scenarios, different combinations have to be experimented one by one.**
> We do not claim that the combination we found to be optimal for audio-visual data would also directly apply to other domains. However, the general enumeration of combinations can be applied again in other scenarios, allowing one to find the optimal combination there too.
> While GDTs can provide a general framework for expressing and searching for optimal configurations in a wide variety of different scenarios, it is true that in general a given combination is only optimal for one scenario at a time. However, this is similar to any kind of empirical research, where methods are developed and tested against specific datasets: there is no guarantee that the specific finding (e.g. which data augmentation to use) would transfer to other problem variants, but in practice this happens often.
>
> **The experiment is done on a very specific scenario: audio-visual task, from which I believe that the main contribution of this work is more of the improvement of a specific audio-visual self-sup method, rather than a generalized formulation of the transformations.**
> We use audio-visual learning to illustrate the importance of being intentional with the choice of data transformations, and the signal for learning. Indeed, audio-visual representation learning is an ideal domain to test our GDT framework because the space of transformations is much larger in videos, due to the time and modality dimensions, where different transformation properties such as variance to time shifts might be more optimal than invariances. Another contribution of this work are strong learned representations, but these are built on our GDT formulation.

---

### Official Review · AnonReviewer2 · 2020-10-30
**Review for Multi-modal Self-Supervision from Generalized Data Transformations**

**Rating:** 7
**Confidence:** 4

**Review:**

This paper is well written and the motivation is simple and clear. The conclusions are supported by sufficient experiments on various datasets and tasks, e.g.,  video and audio classification and retrieval tasks on datasets such as HMDB-51, UCF-101, DCASE2014, ESC-50 and VGG-Sound. The self-supervised pretraining is conducted on VGGSound, AudioSet, and IG65, showing the benefits of GDT in multiple source datasets. The introduced Generalized Data Transformations could potentially benefit multi-modal self-supervised learning in incorporating more transformations.

I have the following comments:

1. In Section 3.1, the authors discussed contrastive audio-visual self-supervision. Can the model generalize to other supervision signals? If so, what's the limitation of the selection process generalization to more self-supervised signals?

2. This paper focuses on video and audio learning. How can this framework be generalized to video and text? The authors may share some insights in the conclusion.

3. Will the order of the hierarchical transformation affect the feature learning process? For example, the order of t1 and t2 is swapped in Fig. 1 A.

---

> ### Author Response · Authors · 2020-11-13
> **Response to Reviewer 2**
>
> We would like to thank Reviewer 2 for their time and constructive comments and appreciation for the various tasks, datasets and benchmarks we have tested.
>
> **1. In Section 3.1, the authors discussed contrastive audio-visual self-supervision. Can the model generalize to other supervision signals? If so, what's the limitation of the selection process generalization to more self-supervised signals?**
> While contrastive audio-visual self-supervision is a large and growing field on its own, our framework  can also be applied to supervised learning: In particular, a recent work, [supervised constrastive learning](https://arxiv.org/abs/2004.11362), show how supervised learning can be formulated as a contrastive task. Our framework could be applied to incorporate further desirable invariances and (what would be more novel) variances/distinctiveness. Furthermore, the transformations that we have introduced, such as cross-modal time-reversal, can also be applied to any other audio-visual learning method, not just contrastive or self-supervised ones.
>
> **2. This paper focuses on video and audio learning. How can this framework be generalized to video and text? The authors may share some insights in the conclusion.**
> Indeed, our framework can be applied to other domains, such as video-text. Given any novel input domain, such as text, one needs to first determine the set of data transformations that can be applied. An overview is given e.g. [here](https://amitness.com/2020/05/data-augmentation-for-nlp/). In our updated manuscript, we will add a paragraph on how to apply GDT with modalities other than audio-video.
>
>
> **3. Will the order of the hierarchical transformation affect the feature learning process? For example, the order of t1 and t2 is swapped in Fig. 1 A.**
> For most of the transformations we introduce, the order does matter. Otherwise inconsistent learning signals can be constructed, and that is why our framework relies on a hierarchy (see Fig. 1a). We give more details into the theory of consistent vs. consistent combinations in Appendix A.1. There are only a few cases where the order is not important, for operations that commute, e.g. whether one first takes a temporal crop and then augments a video or the other way around.

---

### Author Response · Authors · 2020-11-21
**Revised Version: more experiments/ablations**

We thank all reviewers for their constructive feedback and insightful comments.
At this stage, we have finished our revised version of the paper that we believe has addressed all comments raised and now also includes:


1. **Ablation results when using 3 variances** as ideated by R4. As shown in the updated Table 1, it indeed gives a boost in finetuning performance (58.2 vs 60.0), but a slight decrease in retrieval (50.2 vs 47.8). The decrease in retrieval might be due to the 3-variance model becoming more tailored to the pretraining dataset and, while still generalizeable, its frozen features have a slightly higher domain gap compared to the downstream dataset.
2. **Linear probing results** (R4): We achieve a performance of 55.1%, 85.3% for HMDB-51 and UCF-101 respectively (e.g. compared to 54.8%, 83.4% of the MIL-NCE model) and have added this to the appendix Tab. 9.
3. **Intuition behind video transformation** (R1). We have added another paragraph describing some intuitions behind the learning signals provided by the transformations and their invariant vs variant version.
4. **Description of using GDT for video-text** learning (R2, R3), which we have added in the Appendix A2.1.
5. **Better description of how GDT can be used** in the context of searching for optimal combinations (R3), which we have elaborated on in the introduction and conclusion.
6. **More clear explanation why audio-visual** learning is the focus of this paper (R2, R3). As noted in the responses to R2 and R3, we've added details in the methods section on why we explore the audio-visual case in particular in this paper (many possible transformations, additional dimension of time, no clear concensus on invariance or variance being better).
7. **Differentiation to CMC** (R1). We have added a detailed differentiation to the CMC paper in our related works section, as described in our response to R1.
8. **Citation bug** of SimCLR (R1). We have fixed this and checked all other references.
9. **Fixed minor formatting** issue in first revised version (R1).

---

> ### Comment · AnonReviewer4 · 2020-11-22
> **Precision about point 1.**
>
> Thanks for updating the paper.
>
> Regarding the 3 variances point, in the paper it is said *row (m), which shows the (DS, TR, TS)-variance case, yields the best performance and improves*. However this seems to be only true for the FineTuning performance as explained by the authors (and not the retrieval metric).
>
> In particular now in 4.2. when it is said *Given our best learning setup from Sec. 4.1*, what is the best learning setup in that case? As I see that the numbers stayed the same in 4.2. I am assuming that only 2 variances were used. I wonder if there is a plan to use 3 variances for the final experiments? Would that lead to even better final performance in Table 4.? If not, it would be worth clarifying and explaining why only 2 variances were finally chosen as it is now unclear what is the "best setup" in the ablation 4.1.
>
> Finally in Table 4, it seems that the number for GDT should not be in bold for UCF since XDC does better (95.5) when trained on IG65M. I am guessing it should be underlined? Please fix the Table in the final version.

---

> > ### Author Response · Authors · 2020-11-22
> > **Clarification on point 1 for Reviewer 4**
> >
> > Thank you for spotting this. We will adjust our wording in the paper to more accurately describe (row m), as we did in the response earlier and fix the underlining vs bolding mixup of the 95.2% and 95.5% numbers.
> >
> > Regarding the 3-variances case, we will aim to rerun our experiments for the final version as these take quite some time (Audioset and IG65M are quite big datasets). For now, we have adjusted the wording such that we are taking the second best setup (row l).

---

### Author Response · Authors · 2020-11-23
**Updated version + summary of changes since submission**

We would like to again thank the reviewers for their positive feedback and thoughtful suggestions to improve our submission.
We have worked on incorporating both initial and follow-up suggestions in the discussion period and have already uploaded a [first revised version](https://openreview.net/forum?id=mgVbI13p96&noteId=TynXXpX6NM). We have now uploaded our second revised version with further changes which include updated references and slight reformulations.

In summary, compared to our initial submission, we have:
* Added a more comprehensive motivation on why we focus on audio-visual learning instead of images (page 5)
* Elaborated on GDTs' Applicability:
    * Added a description on how GDT could be used for searching for optimal transformation combinations (page 2)
    * Described how it could be used in the other multi-modal settings, such  as video-text (page 15).
* Added intuition behind the audio-visual transformation (page 7).
* Updated the related works section (differentiation to CMC, added concurrent works MMV, CoCLR).
* Added a new experiment evaluating 3-variances (Tab. 1, row (m)).
* Additional linear evalution results for HMDB-51 and UCF-101 (Tab. A3).

---

### Decision · Program_Chairs · 2021-01-07
**Final Decision**

**Decision:**

Reject

**Comment:**

The authors provided a comprehensive rebuttal to the reviewers' feedback that addressed most of the concerns. AnonReviewer3 raised some major concerns that were partially resolved in a revision. The paper has received a split recommendation from the reviewers but within the review and discussion periods, there was no strong support towards accepting the paper. Although the paper has received some positive feedback, some of the reviewers' concerns were not fully addressed. I'd recommend the authors to address all the comments and add clarifying notes to the paper to avoid such misunderstandings if they decide to resubmit the paper to another venue.